# Self-Induced Internal Corrosion Stress Transgranular Cracking in Gradient-Structural Ploycrystalline Materials at High Temperature

**Xianjun Lei [1,2], Xiaopeng Wang [3,4], Fantao Kong [1,2,*], Haitao Zhou [5] and Yuyong Chen [1,2]**

[1]  State Key Laboratory of Advanced Welding and Joining, Harbin Institute of Technology, Harbin 150001, China; lxjhit2018@163.com (X.L.); yychen@hit.edu.cn (Y.C.)
[2]  Department of Materials Science and Engineering, Harbin Institute of Technology, Harbin 150001, China
[3]  Center of Analysis and Measurement, Harbin Institute of Technology, Harbin 150001, China; wangxiaopeng@hit.edu.cn
[4]  National Key Laboratory for Precision Hot Processing of Metals, Harbin Institute of Technology, Harbin 150001, China
[5]  Shanghai Spaceflight Precision Machinery Institute, Shanghai 201600, China; htzzz0313@163.com
[*]  Correspondence: kft@hit.edu.cn; Tel./Fax: +86-0451-86418802

**Abstract:** Self-induced internal corrosion stress transgranular cracking is investigated theoretically and experimentally linking grain boundary wetting (GBW) and grain boundary diffusion (GBD) to improve the ability to reveal the micro mechanism of crack in compositional gradient-structural intermetallic materials. Theoretical analysis shows that the grain boundary wetting and diffusion induce the diffusion-coupled dynamic internal stresses, and their interaction leads to crack nucleation. The experimental results show a stress concentration zone have been established at the grain boundary interface where the cracks preferentially nucleate and then extend through the inside of the grain to both sides, forming a typical transgranular fracture.

**Keywords:** corrosion stress; transgranular fracture; grain boundary wetting; grain boundary diffusion

## 1. Introduction

The fracture phenomena in polycrystalline metals have inspired renewed interest in the mechanism of cracking when solid metals are in contact with liquid metals. Especially, in some material-processing scenarios, such as joint processes [1–4], galvanizing [5–7], heat treatment [8], smelting and solidification [9–11], hot working [12], nuclear industrial [13], microelectronics, data storage technology [14], corrosion science [15–17] etc. The understanding of the fracture mechanism will provide useful insights into the strengthening mechanisms of materials with novel structures or new perspectives on designing materials. Usually, this striking and interesting catastrophic intergranular failure is usually known as liquid metal embrittlement (LME) [18] and stress corrosion cracking (SCC) [19]. As of today, these two ways reducing the mechanical properties of structural materials ways are studied separately. A nanometer thick liquid metal film along the grain boundary was taken as one of the significantly important characteristics for embrittlement in many typical LME systems [20–23]. SCC is a characteristically complicated process and subtle to some processes which create stress concentrators combined with the critical local solution chemistry for cracking [16,19,24–28]. Mechanically and morphologically, two kinds of fracture result in the propagation of cracks in presence of liquid metal and stress or metallic or nonmetallic ions and stress [29], and a large number of cracks grow along intergranular film or notch and intergranular fracture is presented.

Studies aforementioned were mainly involved in the intergranular crack while the dynamic transgranular corrosion crack gained less attention and no intermetallic compound (IMC) formed at interface. IMCs, which are composed of long columnar grains and normal

to a composition-gradient interface, can be formed at the interface in some experiments of liquid metals contacts with solid metals [30–34]. Usually, the formed intermetallic will improve the corrosion resistance and prevent corrosion fracture, yet cracks parallel to the interface can be found. This is very different from the general LME and SCC cracks. The direct contact between the solid metal and the liquid metal is detached by an intermetallic film and the preliminary contact is removed, and the LME does not occur [18,35]. It is likely that the cracking tendency in IMC layer is dependent on the composition of alloy, because of variation in diffusion characteristics such as the magnitude of the composition range and the interfacial curvature variation between the liquidus and solid. The phenomenon is not understood hitherto even qualitatively. On the other hand, previous research was mainly focused on cracking performance on a macro-scale. The simple reason given is that the excessive local residual stress developed in such intermetallic compounds film but no detail and essential causes about the internal stress. The important role played by residual stresses in the posterior crack behavior of the material is now not well known. Lack of in-depth understanding of the root causes of cracking will be an impediment to designing engineering parts for safety-critical applications.

The stress-assisted embrittlement is closely related to chemical interdiffusion driven by the chemical potential linking to the synergistic effects of impurity diffusion motivated by stress gradient and moving of GB [36–39]. Otherwise, a significant stress may generate in oxide film on account of the thermal protective materials during oxidation or the scale growth for superalloy [40–42]. Residual stresses may be emerged mainly in the proximity of surface in metallic materials, and cracks will, in a number of instances, be originated in the centers of grains and out of any neighboring GBs [43]. Consequently, the local stress state is pertinent to the cracks nucleation near the surface, and it can even have an effect on the growth of macroscopic (engineering) cracks. GB cracking controlled by surface and grain boundary diffusion (GBD) have been also reported [44–46]. What's more, crack occurs primarily by intergranular at hardly applied stress levels [47,48]. The crack formation was evidently linked with diffusion of melt into GB, causing the mechanical stresses on account of the atomic volumes of difference between matrix and diffusant [22]. All GB migration, which was taken as a significant component to the evolution of polycrystalline microstructures, necessarily creates mechanical stresses/strains [49]. The reasons for this are multifaceted, but, one of the contributing factors for intergranular cracking may be stress normal to the GB [50], the normal stress will be generated during the GBD process resulting from the dependence of grain boundary energy on composition [51]. These results indicate clearly the crucial role played by the stress state of the GB which may be useful in engineering, but they do not give any insights into the microscopic mechanisms and experimental evidence of crack growth.

In this paper, based on the systematic metallurgical interaction between the IMC layers formed in situ at interface and TiAl melt, the micro causes of the transgranular crack nucleation and growth of cracks coupling with GB wetting and diffusion are investigated from theoretical and experimental. The dynamic evolution of internal stress coupled with GB wetting and diffusion is analyzed in theory firstly, introducing a framework to comprehend the internal stresses coupled with diffusion in affecting the posterior crack behavior of the material by focused considering matter flow in dynamic grooves. Here the emphasis is on the stresses interacting at grain boundaries, as they are probably the most relevant in the context of self-induced internal corrosion stress transgranular cracking (SICSTC). The mechanism of transgranular cracks paralleling to the interface is obtained by analyzing the evolution of internal stress coupled with the GBD in grooves under different corrosion time, and a crack mechanism map has been revealed. It provides a deep understanding of the root causes of cracking and pave the way for developing new engineering technologies, such as surface modification, a joining process and smelting technology.

## 2. Experiment

The Ti-47Al (at.%) alloy samples were supplied in the form of a cylindrical cast ingot which was produced by induction skull melting (ISM) under an argon atmosphere, with the raw materials were shown in Table 1. To promote uniformity, the ingot is remelted at least five times. The cast ingot had exact chemical compositions of Ti-47.05Al (at.%) by EPMA analysis. The ingot was cut into bars with size of 8 mm in diameter and 90 mm in length by wire electrode cutting, and the oxide on the surface were removed to meet the experimental requirement by abrasive paper. Placing each bar into a Nb container (Table 1) with external diameter 12 mm, inner diameter 8 mm and surface finish 1.6. The Nb container was fixed in the setup such as in Figure 1, and vacuum to $5 \times 10^{-3}$ Pa, then filled with high-purity argon (0.5 MPa) as a protection. At a heating rate of 50 °C/min to 1600 °C and keeping different times, the samples were quenched quickly, the solidified bars were cut transversely and then ground by 60–2000 grit papers, polished using diamond solutions (2.5 μm and 1 μm) and finally polished with a 0.04 μm colloidal silica suspension. A ZEISS scanning electron microscope (SEM) (Carl Zeiss, Merin compact, Germany) equipped with an Aztec Energy EDX (Oxford Instruments, UK) was employed to examine the microstructure and the major elements concentration of the samples. The electron probe microanalysis (EPMA) microscope (JEOL JXA-8230, Japan) with a working distance of 11 mm at 20 kV was used to measure the elemental distribution of the samples.

**Table 1.** Component and form of materials used in experiments.

| Material | Form | Chemical Composition (wt%) | | | | | | | | | | | | |
|---|---|---|---|---|---|---|---|---|---|---|---|---|---|---|
| Ti (bala.) | Sponge | Fe 0.015 | Si 0.009 | Cl 0.047 | C 0.009 | N 0.005 | O 0.047 | Mn 0.003 | Mg 0.004 | H 0.001 | | | | |
| Al(bala.) | Lump | Fe 0.1 | Si 0.03 | Ga 0.02 | Cu 0 | Mg 0 | Zn 0.01 | | | | | | | |
| Nb [a] (bala.) | Tube | C 8 | N 49 | H 3 | O 110 | Fe 15 | Si 30 | Mo 32 | W 140 | Ti <5 | Ta 500 | Ni <5 | Hf 20 | Zr 32 |

[a] Chemical composition ≤ (ppm wt).

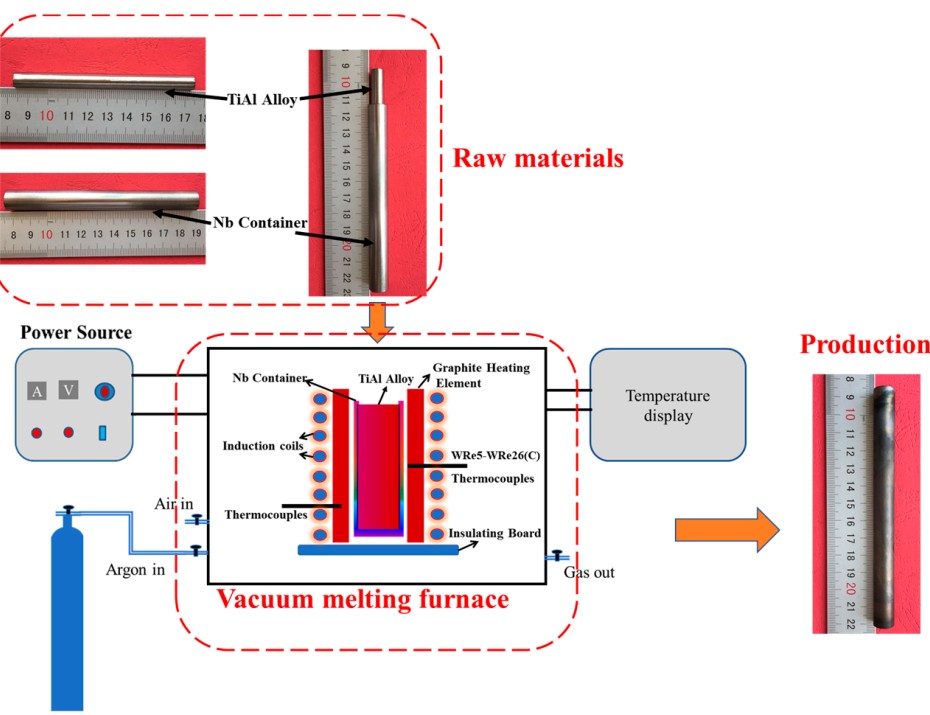

**Figure 1.** Sketch of experimental apparatus [52] (with the permission from Ref. [52]).

## 3. Results

In order to figure out the characteristics of cracking, the representative microstructure of the fracture surface from the corrosion samples were examined by scanning electron microscope (SEM). Figure 2 shows the crack morphology of the interface subjected to SICSTC, and cracks are mainly found in $\varepsilon+O_2$-IMC layer Figure 2a,b and $\delta+\beta$-IMC layer Figure 2c–f, and the corresponding chemical formula revealed in Table 2. Figure 2a shows the crack appearance of the V-grooves suffered from corrosion stress, in which it can be found that the crack propagates through the columnar IMC grains continuously and no obvious crack branches can be detected which is a typical characteristic of transgranular crack, with the direction of crack propagation paralleling to the interface. Moreover, the continuous growth of transgranular cracks was located in two thirds of the V-grooves depth and parallel to the interface near the groove tip. It is interesting to note that transgranular cracks are clearly visible at the groove boundaries and ambiguous inside them, implying the transgranular cracks are preferentially initiated at surface and propagated across the GB. In additional, the cracks seem to joint to a straight line. For 0.5 h corrosion, obvious wetting channels appear along the grain boundaries in columnar $\varepsilon+O_2$-IMC grains. The crack formed at 0.8 times of the wetting channel length and no branches, which is perpendicular to the wetting channel and parallel to the interface, demonstrating an obvious transgranular crack. It is interesting to note that the crack is wider at grain boundaries than in columnar grains, indicating that the crack nucleates at the grain boundary and extends across the interface again. Compared with Figure 2a, the crack has a bigger width and more unambiguous. In contrast, the cracks traverse the wetting channels which are not vertical to the interface show a slight deflection and seem to have a tendency to be normal to the wetting channel, with the width of the cracks is smaller, as shown in Figure 2b. Else, there seems to be a microcrack perpendicular to the wetting channel boundary which is not vertical to the interface. As the $\delta+\beta$-IMC layer is corroded, the location of the crack gradually shifts towards the Nb container and cuts though the bottom of U-grooves, as depicted in Figure 2c. It can be clear to see that the continuous crack which traverses the hump-like $\delta+\beta$-IMC grains propagating parallel to the interface and no crack branching is detected. Which is accordance with Figure 2a,b. It is note-worthy that the crack is perpendicular to the intersection where the flat wall meets the curved boundary of U-groove tip after a slight deflection during its extension. The crack is more pronounced at the edge of the U-groove root compared with Figure 2a,b. It can be inferred that the crack nucleated at the interface. For 1.5 h corrosion, the crack can be found at the bottom of the widened U-grooves and traverses the new formed concaves, propagating across the hump-like $\delta+\beta$-IMC columnar grain and paralleling to the interface, as disclosed in Figure 2d. It is worth to note that a slight crack deflection is detected in the U-groove, the crack is more obvious near the boundary with the advent of the concave. Experimental evidence cited here, as well as described in Figure 2c, supports a contention that the crack originates the boundary. When the corrosion time reaches two hours, a significant crack extends along the bottom of the U-groove and traverses through the columnar IMC protuberances, indicating an explicit transgranular crack. It is essential to uncover that the crack originates from the boundary on account of the width of the crack at the interface is wider than that within the grain, as portrayed in Figure 2e. It is note-worthy that two new V-grooves seem to be formed at the bottom of the U-groove and the crack goes though the root of the two new V-grooves. By comparing the results of the crack width near the boundary and inside the groove, the crack originates the boundary is proposed. After 2.5 h for corrosion, Figure 2f shows that the crack extends along the bottom of the broadened U-groove and across the slender columnar IMC grain. Compared with Figure 2e, the width of the crack seems to increase. Cracks grow through both phases and no preferable path, indicating no remarkable toughness difference between $\delta$ and $\sigma$ phase. Additionally, a sheet coating seems to have formed on the bottom of the broadened U-groove, which necessarily accompany the formation of cracks, reducing the rate of crack formation and increasing times to failure. This is coherent with Ainslie's concept [53].

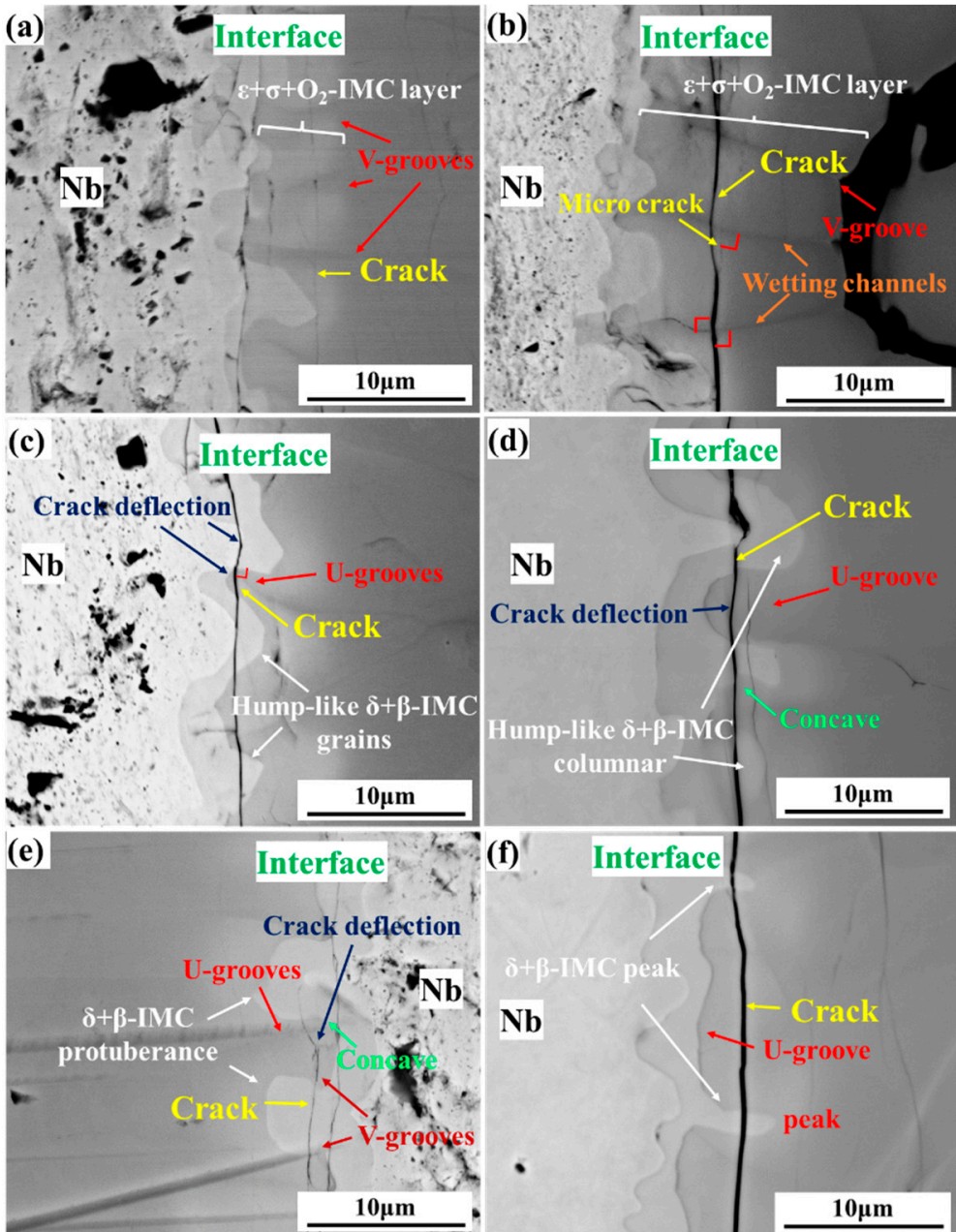

**Figure 2.** SEM images of interfacial crack along with interface evolution for different corrosion time at 1600 °C: (**a**) 0.2 h, (**b**) 0.5 h, (**c**) 1 h, (**d**) 1.5 h, (**e**) 2 h, (**f**) 2.5.

**Table 2.** The phases and the correlative chemical formula.

| Phase | Chemistry Formula |
|---|---|
| δ | $Nb_3Al$ |
| σ | $Nb_2Al$ |
| ε | $(Ti_{1-x}Nb_x)Al_3$, $TiAl_3(h)$, $NbAl_3$ |
| $O_2$ | $Ti_2NbAl$ |
| β | (Nb)ss |

Experimentally, it is clear from microstructure observations on fracture pattern that the path of crack is predominantly transgranular, though the morphology of grooves changes with the corrosion time. It can be concluded that the original crack core is a cleavage

crack at the GB interface which traverses two neighboring columnar IMC grains and, upon meeting another cleavage crack at the center of IMC grains, interconnecting joint thereafter.

Fracture surface of a sample have exhibited a cleavage crack morphology suggesting the presence of a continuous transgranular crack along the interface, as shown in Figure 3. Figure 3a shows a smooth macro fracture surface between the Nb container and the TiAl melt. Figure 3b,c show the morphology of crack surface inside the Nb container and the surface of TiAl sample, respectively, demonstrating a cleavage crack with the river pattern morphology.

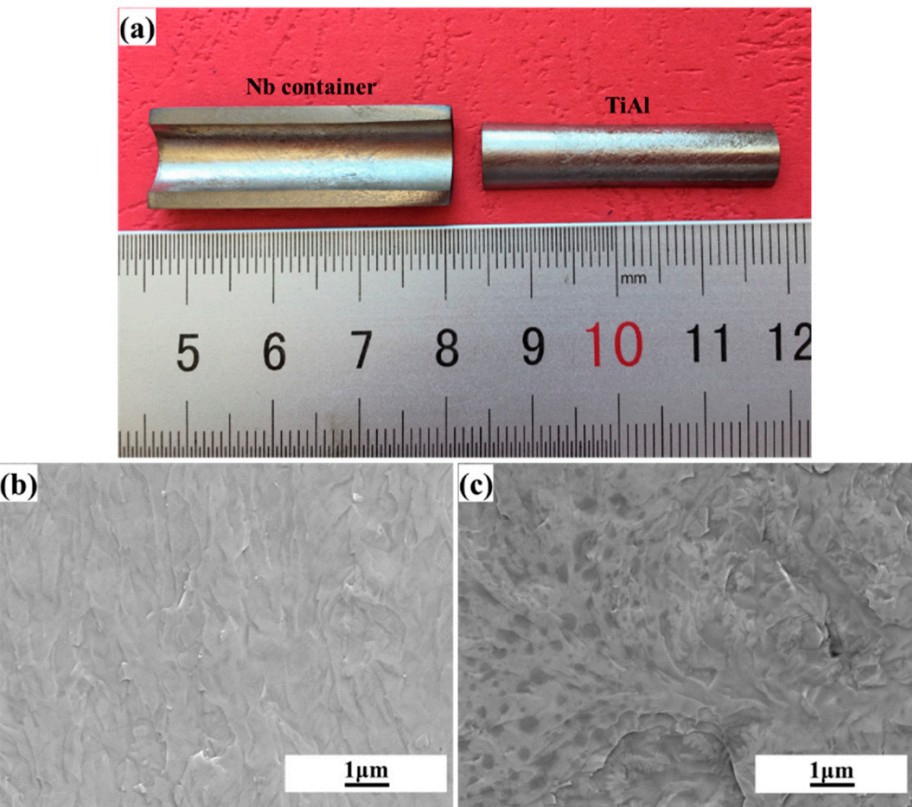

**Figure 3.** Fracture morphology of the sample: (**a**) Macro fracture picture of the interface, (**b**) SEM-SE micrograph of crack surface inside the Nb container, (**c**) SEM-SE micrograph of crack on the surface of TiAl sample.

## 4. Discussion

### 4.1. The Source of the Force

Based on the analysis in Section 3, the crack is always transgranular while its location changes with the evolution of interface morphology, which is very different from other SCC. Based on the previous study [54], three IMC layers consisting of many small and parallel columnar crystals with a certain surface curvature formed on the surface of Nb container, as shown in Figure 4. This interfacial configuration has instability, which is coincident and consistent with other studies [55,56]. The macroscopic behavior is that the TiAl melt corrodes the Nb container in an immersive manner. The root of the GB propagates in a diffusion regime and widens in a dissolution regime, which is inferred to experience a liquid grooving [57]. Thermodynamically, the morphology of the GB grooves will evolve spontaneously (from V-groove to U-groove) [58]. Therefore, a systematic parametric study is carried out in this section to investigate the factors affecting the transgranular cracks.

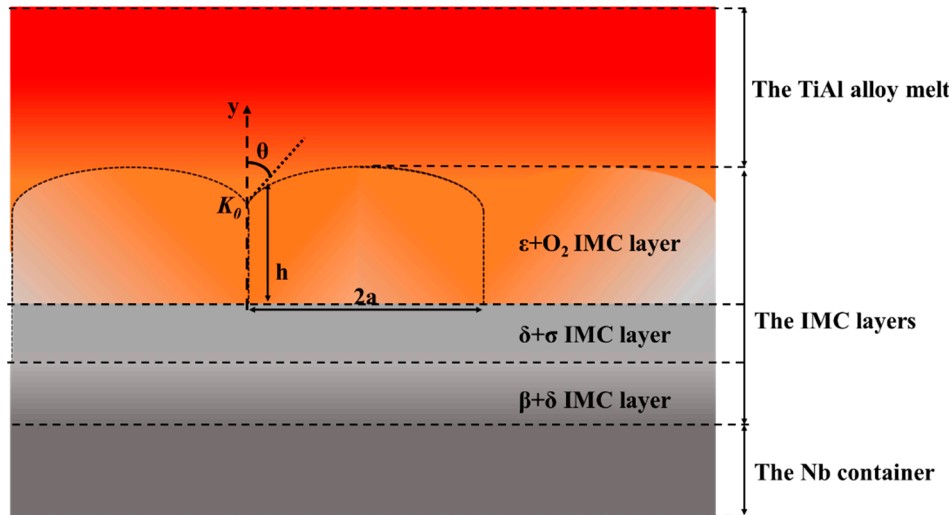

**Figure 4.** The geometry schematic of interface when the TiAl melt corrading the Nb container.

Fluctuations can occur at grain boundaries when the TiAl melt corrading the Nb container, as shown in Figure 4. Columnar IMC grains with even length 2a and height h perpendiculars to the substrate [52]. It is assumed that this initial configuration is one of equilibrium with the same uniform curvature on the surface. This curvature must be given by:

$$K_0 = \frac{\cos \theta}{2a} \tag{1}$$

where $\theta$ is the dihedral angle at the intersection of the GBs, and the concave surfaces are negative. Therefore, the residual stress in the groove root can be obtained by:

$$\sigma_0 = \gamma_s(\sin \theta / h + K_0) \tag{2}$$

where $\gamma_s$ is the surface tension of the related IMC layer.

With the advent of a concentration gradient, the diffusive flux toward the region of maximum stress can be driven by the chemical potential [41]:

$$\nabla \mu = -\Omega \nabla \sigma \tag{3}$$

where $\nabla \sigma$ is the relevant stress gradient.

When the IMC layer is in contact with the TiAl alloy liquid, The Equation (3) can be concretized as:

$$\mu_B = \mu^0 - \sigma(y)\Omega \tag{4}$$

$$\mu_S = \mu^0 - K(s)\gamma_s\Omega \tag{5}$$

where the subscripts $B$ and $S$ represent the boundary and surface quantities, respectively, y and s are coordinates along the boundary and the surface. $\mu^0$ is a standard potential, $\sigma(y)$ is the local stress normal to the boundary, $\Omega$ is the atomic volume, and $K(s)$ is the local surface curvature.

Under the chemical potential, atoms would move into the GB from the interface, and the flux of atoms, $J$, is given by the Nernst-Einstein equation:

$$J_B(y) = -\frac{\delta_B D_B}{\Omega \kappa T}\left[\frac{d\mu_B(y)}{dy}\right] \tag{6}$$

$$J_B(S) = -\frac{\delta_S D_S}{\Omega \kappa T}\left[\frac{dK(S)}{dS}\right] \tag{7}$$

where $\delta_B D_B$ and $\delta_s D_s$ signify the diffusivities of the boundary and surface, $k$ is Boltzmann's constant and $T$ is the absolute temperature.

Combining Equation (3) with Equation (4), the fluxes along the GB ($y$ direction) in Figure 5 are given by:

$$J_{Ti+Al}(y) = -\frac{\delta_{Ti+Al}D_{Ti+Al}}{\Omega\kappa T}\left[\frac{d\mu_{Ti+Al}(y)}{dy}\right] = -\frac{\delta_{Ti+Al}D_{Ti+Al}}{\Omega\kappa T}\left[\frac{d\sigma(y)}{dy}\right] \tag{8}$$

$$J_{Nb}(y) = -\frac{\delta_{Nb}D_{Nb}}{\Omega\kappa T}\left[\frac{d\mu_{Nb}(y)}{dy}\right] = -\frac{\delta_{Nb}D_{Nb}}{\Omega\kappa T}\left[\frac{d\sigma_{lb}(y)}{dy}\right] \tag{9}$$

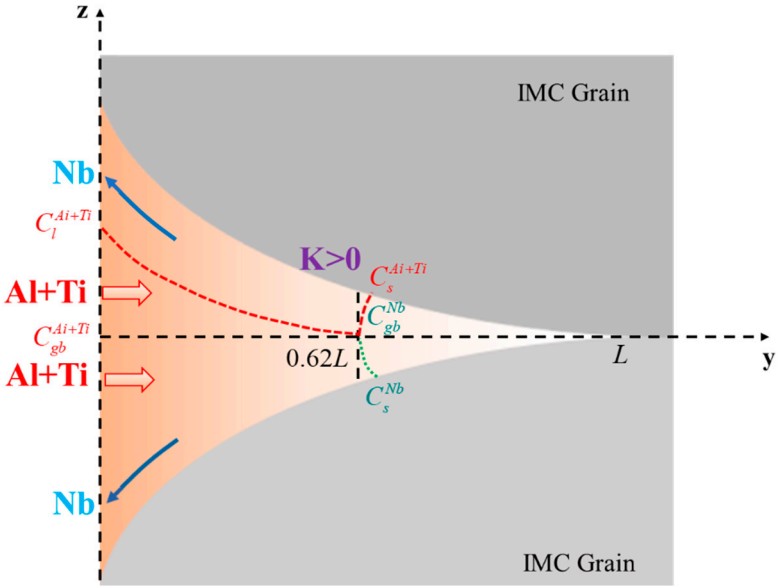

**Figure 5.** Illustration of diffusion of Al, Ti and Nb atoms along the V-type GB in the solid IMC phase. The diffusion concentration in the GB has a subscript gb, and those of in surface with a subscript s and in TiAl melt with a subscript $l$, respectively.

This process is closely related to the GBW because of the diffusion along the prewetted *GB* is fast enough, the stresses inside the prewetted *GB* phase are inclined to a steady state [54]. The corresponding normal stress along the *GB* can be expressed as [14]:

$$\sigma_{GB}(y,t) = \gamma_s k(0) - E^* \int_0^\infty K(y,z)\frac{\partial\omega(z,t)}{\partial z}dz \tag{10}$$

$$\frac{\partial w(y,t)}{\partial t} = \frac{-\delta D_{gb}\Omega}{kT} \cdot \frac{\partial^2\sigma(y,t)}{\partial y^2} \tag{11}$$

where $E^* = \frac{E}{4\pi(1-\nu^2)}$ is the equivalent elastic module, and:

$$K(y,z) = \frac{1}{y-z} + \frac{1}{y+z} - \frac{6y}{(y+z)^2} + \frac{4y^2}{(y+z)^3} \tag{12}$$

The *Nb* dissolution from the *Nb* container is equal to the dissociation of Nb atoms from IMC layer to the IMC layer-TiAl melt interface and the mass transfer of Nb atoms through the adjacent boundary layer to the bulk TiAl melt. Straumal et.al's [58] research indicated that pressure had a significant influence on grain boundary wetting and solute concentration in it. Normal to the GB ($z$ direction), on the other hand, a pressure P (the

interfacial stress) would be in equilibrium with it if a segment of surface with an isolated curvature $K$ based on the Gibbs-Thompson formula:

$$\ln(\frac{p}{p_0}) = K \cdot \frac{\gamma_s \Omega}{kT} \tag{13}$$

where $p_0$ is the vapor pressure when $K = 0$ (a plane surface), $\Omega$ is the molecular volume, $\gamma_s$ is the surface-free energy per unit area, $k$ is Boltzmann's constant and $T$ is the absolute temperature. Therefore, the stress normal to the surface will therefore be coupled with the surface curvature. At the beginning, we can assume that the depth of the groove is only a few atoms distance, Equation (7a) can be approximated:

$$\frac{\Delta p}{p_0} = K \cdot \frac{\gamma_s \Omega}{kT} \tag{14}$$

According to the kinetic theory of gases [59], the flux of atoms emitted by segment of surface with curvature $K$ is:

$$J_N^{Nb} = \frac{\Delta p}{\sqrt{2\pi MkT}} = K \cdot \frac{p_0 \gamma_s \Omega}{(2\pi M)^{\frac{1}{2}} (kT)^{\frac{3}{2}}} \tag{15}$$

where $M$ is the weight of a molecule.

Yost et al. [60] demonstrated that a normal stress during *GB* thermal diffusion given by:

$$\sigma_n(z, t1) = \sigma_0 \left[ 1 + \beta \exp(-\frac{z}{a}) \right] \tag{16}$$

where $\beta = 1$–2 is the stress concentration factor. The distribution of forces was consistent with Klinger et al.'s work [51].

Compared with Figure 5, some differences of diffusion flux can be found in U-type *GB* but flux along the *GB* (y direction), as shown in Figure 6.

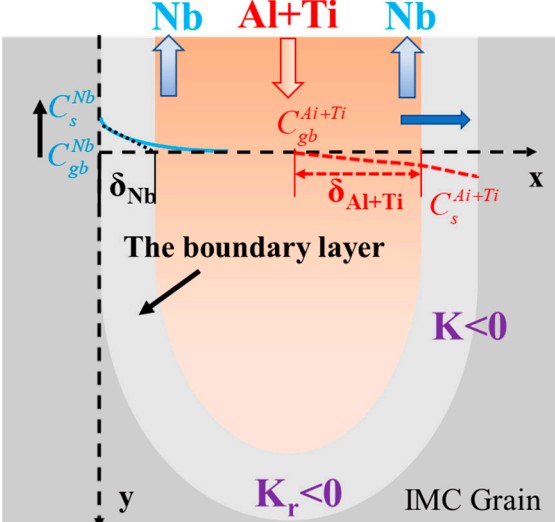

**Figure 6.** Illustration of diffusion of Al, *Ti* and *Nb* atoms along the U-type GB in the solid IMC phase for steady-state dissolution controlled by diffusion. The diffusion concentration in the GB has a subscript gb, and those of in surface with a subscript s and in *TiAl* melt with a subscript *l*, respectively, x: distance from solid surface. δ: effective thickness of diffusion layer.

For a certain point along the flat, free grain boundary surface, the local chemical potential is:

$$\mu(x, t) = \sigma_b(x, t)\Omega \tag{17}$$

and the atom diffusion flux per unit out-of-plane thickness there is:

$$j(x,t) = -\frac{\delta_b D_b}{kT\Omega} \cdot \frac{\partial \mu(x,t)}{\partial x} \tag{18}$$

Combining Equations (9) and (10),

$$j(x,t) = \frac{\delta_b D_b}{kT} \cdot \frac{\partial \sigma_b(x,t)}{\partial x} \tag{19}$$

On the other hand, the diffusion flux is proportional to the gradient of its concentration, it is assumed that the steady-state rate of dissolution is controlled by diffusion.

$$j(x,t) = \frac{D_b}{\delta_b} \cdot (C_i^s - C_i^b) \tag{20}$$

$$\frac{\partial \sigma_b(x,t)}{\partial x} = \frac{kT}{\delta^2}(C_i^s - C_i^b) \tag{21}$$

$$\sigma_b(x,t) = \frac{kT}{\delta}(C_i^s - C_i^b) \tag{22}$$

Therefore, based on Equation (10), the lateral ($x$ direction) flux of elements can be gotten,

$$J_{gb}^{Nb} = -\frac{\delta_{Nb} D_{Nb}}{kT\Omega} \cdot \left[\frac{\partial \mu(x,t)}{\partial x}\right]_{(x=\delta_{Nb})} \tag{23}$$

$$J_s^{Ti+Al} = -\frac{\delta_{Ti+Al} D_{Ti+Al}}{kT\Omega} \cdot \left[\frac{\partial \mu(x,t)}{\partial x}\right]_{(x=\delta_{Ti+Al})} \tag{24}$$

And the lateral stresses caused by diffusion can be obtained according to Equation (14):

$$\sigma_n = \frac{kT}{\delta_{Al+Ti}}(C_{gb}^{Ai+Ti} - C_s^{Ai+Ti}) \tag{25}$$

$$\sigma_b = \frac{kT}{\delta_{Nb}}(C_s^{Nb} - C_{gb}^{Nb}) \tag{26}$$

Similarly, assuming that $\mu_r^{(gb)} = \mu_r^{(s)}$ at groove root, we can acquire:

$$\sigma_r = \gamma_s K_r \tag{27}$$

where $\gamma_s$ is the surface free energy at IMC layer and $K$ is the curvature. In addition, the interfacial vector flux at the groove root,

$$\Delta J = J_r^{gb} - J_r^s \tag{28}$$

where $J_r^{gb}$ means the component into the root and $J_r^s$ out of the root.

Theoretically, self-induced internal corrosion stress is mainly affected by chemical potential gradient, interface curvature variation and interfacial diffusion. Combined with the above analysis, the stress in the V-groove is mainly associated with the chemical potential gradient and the change of the interface curvature, and that of in U-groove is mainly involved in the curvature variation and the GB diffusion.

### 4.2. Crack Mechanism Coupling with Diffusion in Dynamic Grooves

Taking diffusion is driven by chemical potential (concentration) gradients and pressure into consideration according to Section 4.1, the diffusion flux $J$ can be written as:

$$J = -D_i^X(\nabla C_i^X) + J_N^X + \Delta J, [(i = GB, l, S), (X = Al + Ti, Nb)] \tag{29}$$

where $D_i^X$ is the corresponding diffusion coefficient of $X$ in $i$ at temperature $T$, $C_i^X$ is the concentration of $X$ in i.

Based on the Equation (29), the local stresses caused by concentration and pressure gradient play a significant role in the diffusion process from both grain boundaries and adjacent grains. However, it is hard to obtain the related parameters (the pressure factor and diffusivities) in diffusing process. It can be only assessed qualitatively from experimental to understand the relationship between mechanical and material transport, the symmetric boundaries and the identical concentration value at the external boundaries are taken into account in dynamic grooves.

### 4.2.1. Mechanical Interaction and Crack Mechanism in V-Grooves

Figure 7 shows the SEM micrograph of a V-groove (such as in Figure 2a) and SEM-EDS lines scanning of the relevant elements. It can be seen that the penetration of *Ti* and *Al* elements along the GBs is an of intrusive nature accompanying with an obvious curvature $K_1$, resulting in a drift of the neighboring grains away from each other, as shown in Figure 7a. Figure 7c,g reveal that Al and *Ti* elements have positive concentration gradient along line 1, 2, 3 (in Figure 7b), a negative for *Nb*. Hence, $J_l^{Ai+Ti} > J_l^{Nb} > 0$ can be obtained based on Equation (8) and (9), and $\gamma_{GB} > 2\gamma_{lb}$. Therefore, a stress driven diffusion flux of both types of atoms into the GB owing to a large concentration difference at groove interface. Figure 7d,f,h depict that the concentration gradient of *Al*, *Nb*, *Ti* elements can be gotten from the V-groove center to both sides along the lines 4, 5, 6 (in Figure 7b), and the largest concentration gradient is near the crack. A positive concentration gradient for Al and *Ti* elements indicating they diffuse into the boundary and corrode it, that of *Nb* is negative showing it is dissolved. It is logically inferred that $J_s^{Ai+Ti} > 0$ and $J_{gb}^{Nb} > 0$ coupling with $\sigma_n$ and P based on Equations (14) and (16), respectively. It should be noted that a small concentration gradient and short transporting period can be considered at GB surface owing to a relatively small curvature $K_1$, causing a localized corrosion stress. Compared Figure 7d with Figure 7f, it can be found that the center concentration of Al element decreases firstly from line 4 to line 5, and then increases from line 5 to line 6. The biggest concentration of *Ti* occurs at the mouth of the V-groove, then decreasing gradually, while that of *Nb* almost has an opposite trend. This process represents diffusional drift under the condition of $J_l^{Ai+Ti} \gg J_l^{Nb} > 0$ coupling with a large $\gamma_{GB} > 2\gamma_{lb}$. It is concluded that the interaction of internal stress will occur at the boundaries of V-grooves under the driving of chemical gradient or thermal diffusion.

Consequently, the mechanical interaction coupling with GB diffusion can be derived and the stress concentration area can be obtained based on the previous analysis, as shown in Figure 8a. It is logically uncovered the stress concentration area is located in 0.63 depth of the V-groove where the crack is preferentially initiated and traverses the grain boundary and the columnar as illustrated in Figure 8b. This is indirect coincident and consistent with Klinger's opinion, the maximum tensile stress is located in 0.62 depth of the V-groove [51].

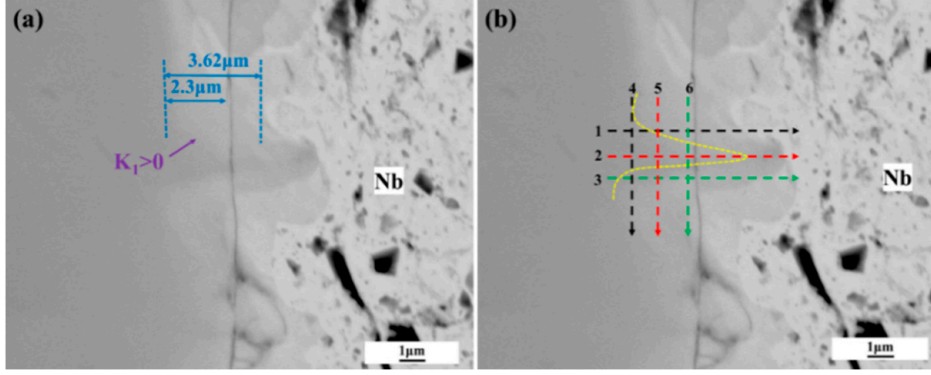

**Figure 7.** *Cont.*

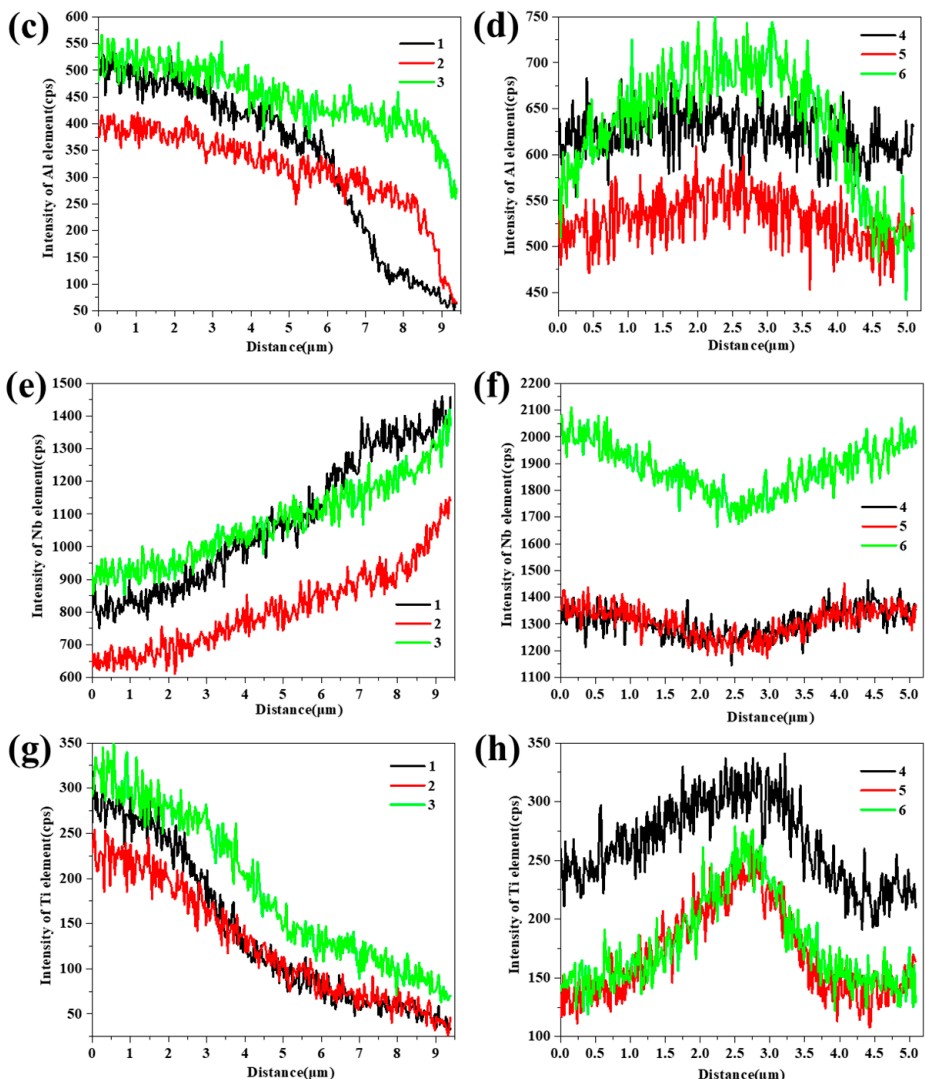

**Figure 7.** SEM micrograph of a V-groove (**a**) and the relevant SEM-EDS line scanning in it (**b**), the relative content of *Al*, *Nb* and *Ti* along the dot line marked in (**b**) by SEM-EDS line scanning: (**c**) and (**d**) for *Al*, (**e**) and (**f**) for *Nb*, (**g**) and (**h**) for *Ti*.

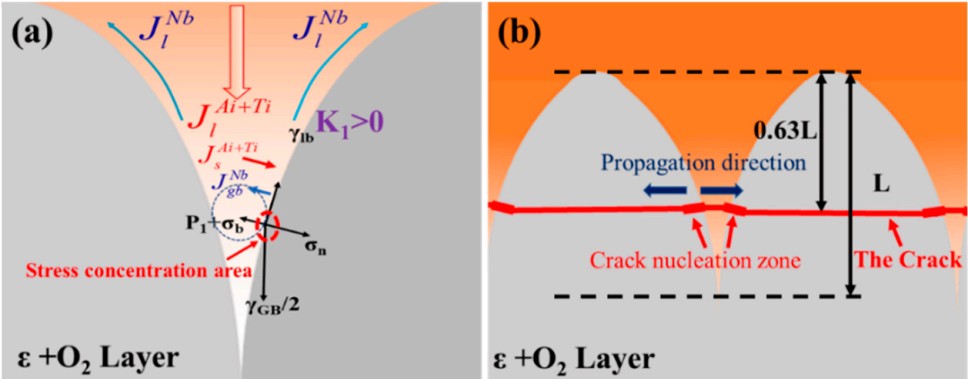

**Figure 8.** (**a**) Schematic illustration of the related mechanical interactions coupled with the mass flux, (**b**) sketch of crack nucleation and propagation.

To elucidate the mechanism which is responsible for the formation of crack in the wetting channels (such as in Figure 2b), EPMA was employed to analyze the map distribution of the relevant elements, as presented in Figure 9. A clear crack traverses through

0.82 times the depth of the wetting channels and columnar δ+σ-IMC grains and elongates paralleling to the interface, as shown in Figure 9a. The crack deflects slightly when it encounters the Nb-rich phase near the wetting channel based on Figure 9a,d $J_l^{Ai+Ti} > J_l^{Nb} > 0$ can be obtained combining with the diffusion of Ti, Al and Nb along the GB based on Equation (8) and (9), indicating diffusional drift under the condition of $J_l^{Ai+Ti} >> J_l^{Nb}$ coupling with a larger, which is similar with that of V-groove. The reason is that Ti, Al atoms diffuse into the Nb container wall (Figure 9b,c), and Nb atoms have an opposite direction (Figure 9d). It is implied that the solid material can be dissolved at penetration front. Otherwise, the two-dimensional compositional contour map across the interface reveals that an overt elements concentration gradient of Al and Nb elements exist while that of Ti elements is ambiguous on account of a low solubility of Ti elements in Nb-Al compounds [61]. This proves that the IMC layers generated in situ at the interface have a component concentration gradient, which is supportive for Figure 4 experimentally again. Based on lateral diffusion of Ti atoms in Figure 9b, $J_s^{Ai+Ti} > 0$ and $J_{gb}^{Nb} > 0$ coupled with σ_n and σ_b, respectively, resulting in widening the groove channels. Otherwise, other experimental studies have shown that the layer thickness of the wetting channel is very small, exhibiting a weak curvature ($K < 0$) with an opening angle at the tip of the penetration front in bicrystal [62], resulting in a pressure P based on Equation (14). Taking an approximate treatment, the wetting angle is wedge-shaped feature, as expected. Therefore, the mechanical interaction coupling with GB diffusion as shown in Figure 10a. The stress level is higher than the prior situation on account of a bigger curvature K₂ at which the GB can hardly endure excessive stresses and crack. It is logically concluded that the stress concentration area is located in 0.82 depth of the wetting channels where the crack is preferentially initiated and traverses the GB and the columnar, as revealed in Figure 10b.

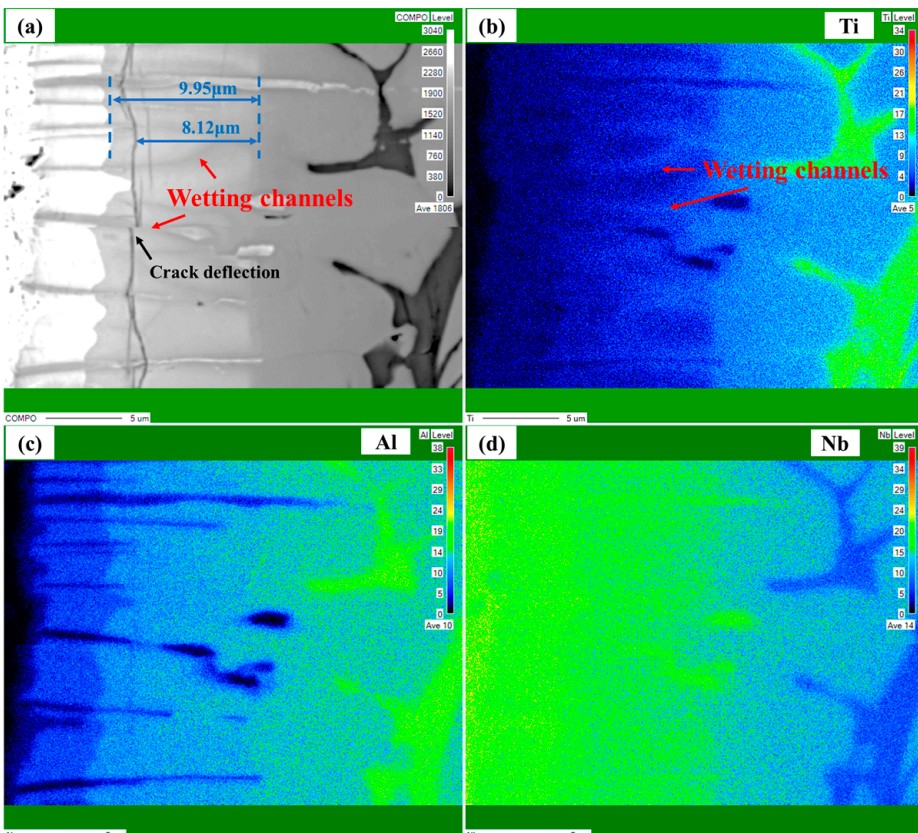

**Figure 9.** The cross-sectional microstructural analysis of the Nb container/TiAl melt interface for 0.5 h holding by EPMA: (**a**) EPMA-BSE image, (**b**–**d**) the spectrum mapping of Ti, Al, Nb in (**a**).

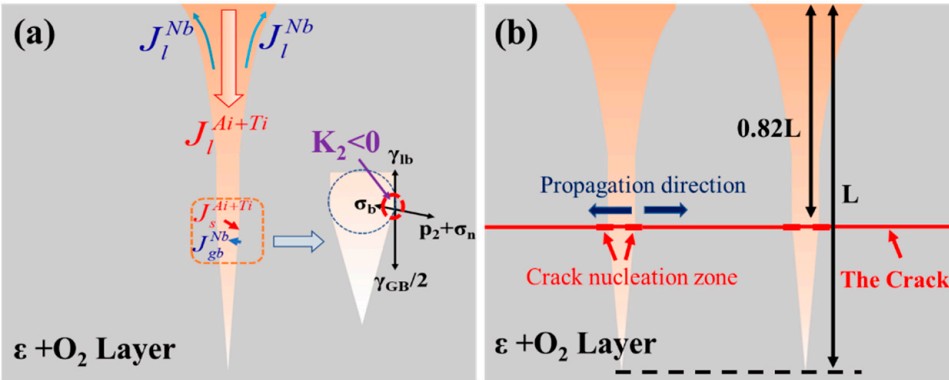

**Figure 10.** (**a**) Schematic illustration of the related stress interactions coupled with element diffusion, (**b**) Schematic illustration of Crack nucleation and propagation.

Based on the previous experimental analysis, it is demonstrated that the nucleation and extension of the crack can be directly related to the depth of the V-grooves.

4.2.2. Mechanical Interaction and Crack Mechanism in U-Grooves

The SEM micrograph of crack in a U-groove (such as in Figure 2c) and the relevant SEM-EDS lines scanning of the relevant elements along the dot lines shown in Figure 11. It can be seen that the transgranular crack without any branches extends continuously between $K_1$ and $K_2$ paralleling to the interface and deflects slightly near the GB, as portrayed in Figure 11a. Otherwise, it is worth to note that the width of the crack at the boundary appears to be wider than that of in IMC grains and interior of the groove, implying the crack source is in the boundary. The concentration of Al, Ti elements deceasing along the GB (Figure 11c,g), respectively, and that of Nb has an opposite trend (Figure 11e), demonstrating $J_l^{Ai+Ti} > J_l^{Nb} > 0$ and $\gamma_{GB} - 2\gamma_{lb} > 0$. It is worthy of noting that the concentration of Nb element along line 1 and line 4 are obviously larger than those of line 2 and line 3, showing that Nb atoms enter the TiAl melt along the interface directly. An obviously decreasing concentration gradient for Ti and Al elements can be obtained during the lines (Figure 11c,g) along the GB, while a stable concentration gradient for Nb element (Figure 11e). It can be logically inferred that a stable boundary layer has been formed near the interface. Additionally, the crack cuts right across the plane where the curvature of GB has a transition, therefore, the crack initiates at points where the curved section ($K = 0$) with two pressures ($P_1$, $P_2$). The lateral concentration gradient still exists at the range of 0.5–2 μm based on Figure 11d,f,h. According to Equations (23) and (24), $J_s^{Ai+Ti} > J_{gb}^{Nb} > 0$, $\sigma_n > 0$, $\sigma_b > 0$. The center concentration of Al element increases firstly from line 5 to line 7, and then decreases from line 7 to line 8, while that of Nb and Ti decreases from line 5 to line 6, then increases suddenly and decreases again by comparing Figure 11d,f,h. It is demonstrated that the concentration gradient of Al, Nb and Ti are the largest in the proximity of the crack where is the stress concentration area. What's more, the concentration of Al and Ti elements along line 5 seems to be higher than those of line 6, revealing $\Delta J > 0$ at the groove root and $\sigma_r > 0$.

The mechanical interaction coupling with GB diffusion can be obtained based on the analysis of Figure 11, as shown in Figure 12a. It is logically induced that the stress concentration area is located in the boundary layer region of $K = 0$ between $K_1$ and $K_2$. Compared with Figure 11, the stress level is higher resulting from GB surface diffusion which results in three different curvatures. Figure 12b reveals that the crack initiates preferentially at the boundary region of $K = 0$ between $K_1$ and $K_2$ and traverses the grain boundary.

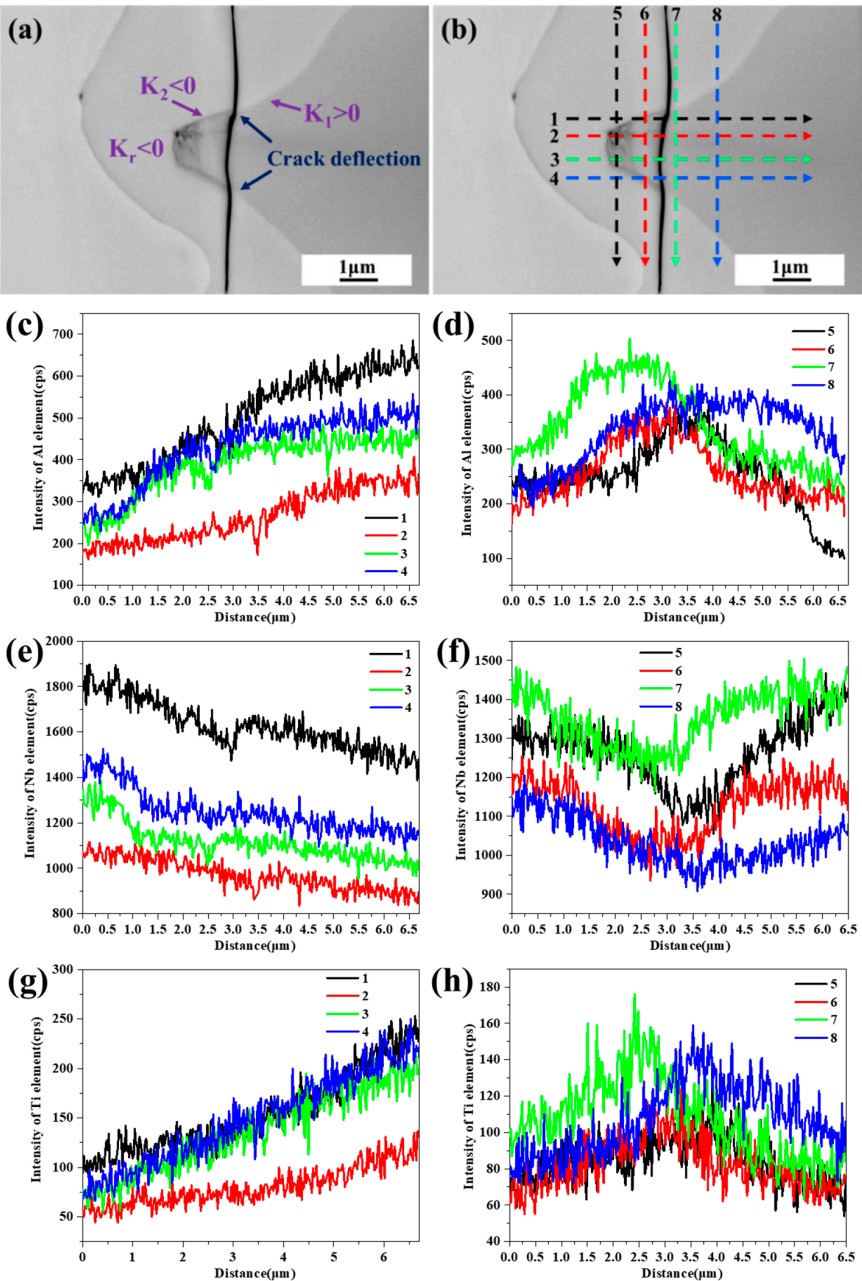

**Figure 11.** SEM micrograph of U-groove (**a**) and the relevant SEM-EDS line scanning in it (**b**), the relative content of Al, Nb and Ti along the dot line in (**b**) by SEM-EDS line scanning: (**c**,**d**) for Al, (**e**) and (**f**) for Nb, (**g**) and (**h**) for Ti.

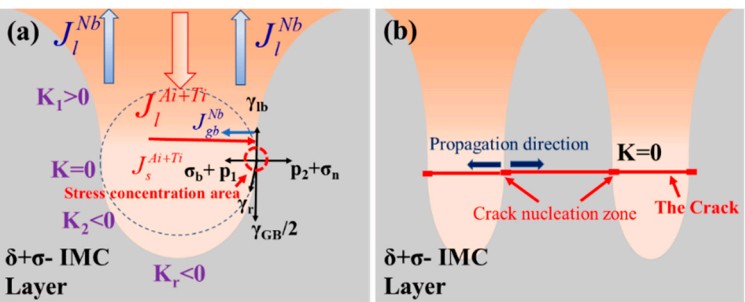

**Figure 12.** (**a**) Schematic illustration of the related stress interactions coupled with element diffusion, (**b**) Schematic illustration of crack nucleation and propagation.

Figure 13 shows the SEM micrograph of U-groove with bigger diameter to depth ratio (such as in Figure 2d) and SEM-EDS lines scanning of the relevant elements. The transgranular crack without any branches extends continuously paralleling to the interface, slightly crack deflections can be found at the GB and inside the groove, as portrayed in Figure 13a. Otherwise, it is note-worthy that the crack has a bigger width at the boundary where the curved section ($K_2 < 0$, $P_2 > 0$) at root meets the flat walls ($K_1 = 0$) than that of in IMC grains and interior of the groove, demonstrating the crack source is located in the boundary and at a point where the curved section ($K_2 < 0$) meet the flat walls ($K_1 = 0$). The concentration of Al, Ti elements deceasing along the GB Figure 13c,g from 1.5 to 0 μm, respectively, and that of Nb has an opposite trend (Figure 13e), indicating $J_l^{Ai+Ti} > J_l^{Nb} > 0$, $\gamma_{GB} - 2\gamma_{lb} > 0$. Compared with Figure 13, the concentration of Al, Ti elements has a slight reduction from 5 to 1.5 μm along the GB, while that of Nb is almost a constant, revealing the U-groove has two flat boundaries paralleling to each other in the range of 1.5 to 5 μm. This confirms $K_1 = 0$ and a stable boundary layer is formed near the flat wall experimentally. In contrast, the lateral concentration gradient still exists at the range of 1.5–2.75 μm based on Figure 13 c, e, g. According to Equations (23), (24) and (26), $J_s^{Ai+Ti} > J_{gb}^{Nb} > 0$, $\sigma_n > 0$. Additionally, the central concentration of Al element increases from line 5 to line 7, while that of Nb and Ti almost has a opposite trend by comparing Figure 13c,e. It is shown that no atomic enrichment zone at the U-groove root, implying $\Delta J \approx 0$ and $\gamma_{GB} - 2\gamma_{lb} > 0$ but less than those of in Figure 11. As per reference [63], the stresses in GB are connected directly with a difference of superfacial tension and that can be estimated by $\Delta\sigma \approx \frac{\Delta\gamma}{a} = \frac{\gamma_b - 2\gamma_{lb}}{a}$ (Where $\alpha$ is the width of GB) which decreases with the increase of $\alpha$. The concentration evolution of Ti element is in accordance with that of Al, as shown in Figure 13g. The mechanical interaction coupling with GB diffusion can be obtained, as shown in Figure 14a. The bigger $K_2$ than in Figure 12, the larger $J_s^{Ai+Ti}$ coupled with a greater $P_2 + \sigma_n$. Therefore, the stress concentration area is located in the curved section ($K_2 < 0$) where the crack initiates preferentially and traverses the grain boundary and the IMC columnar, as presented in Figure 14b. Which matches the experiment perfectly, as shown in Figure 2d or Figure 13a.

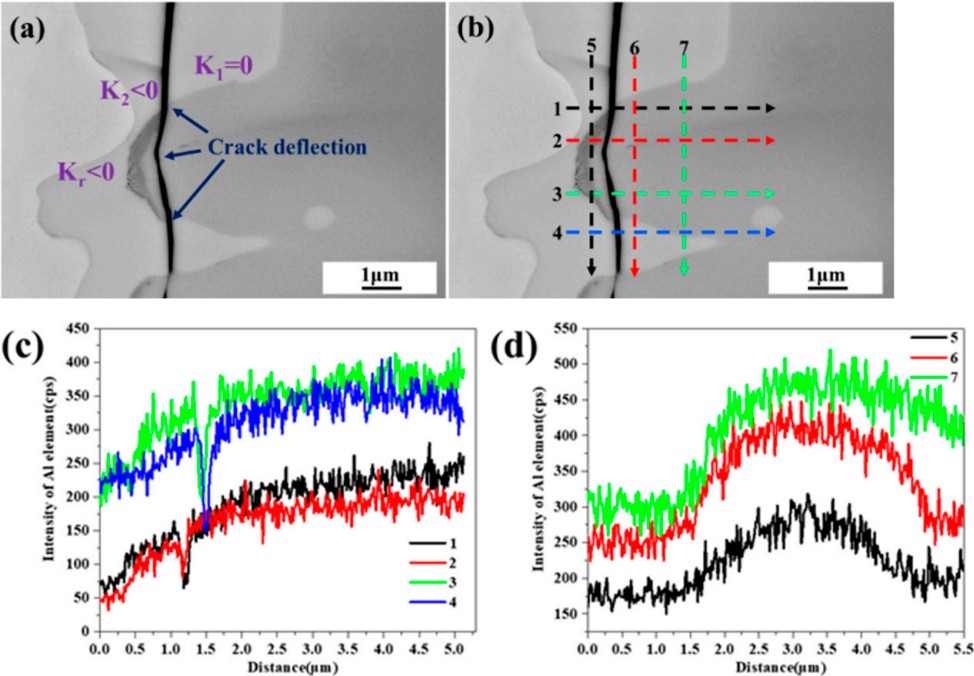

**Figure 13.** *Cont.*

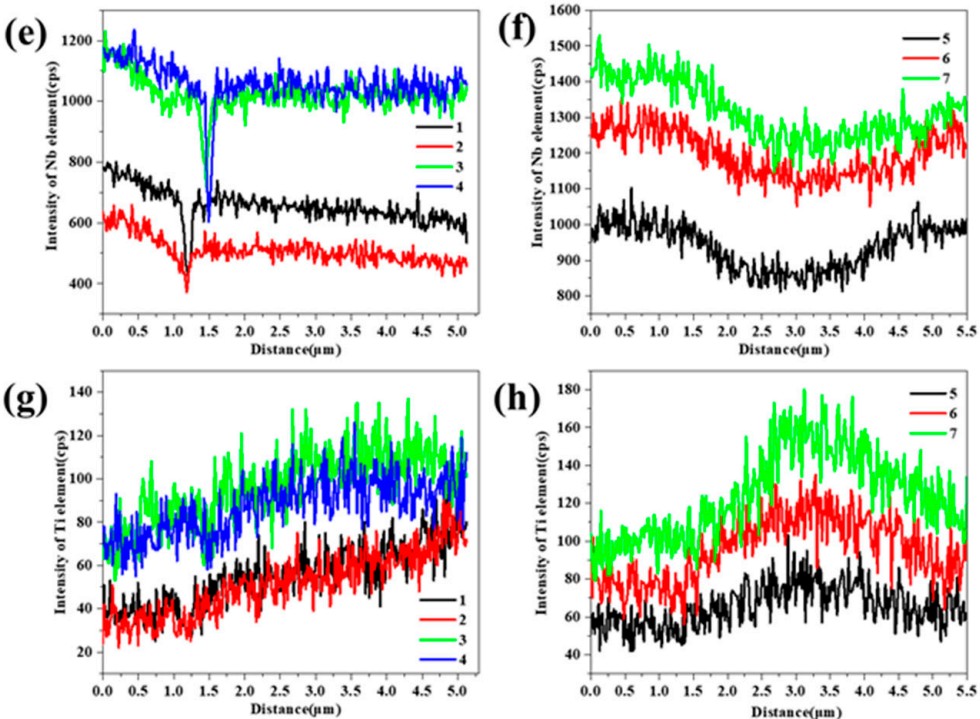

**Figure 13.** SEM micrograph of a U-groove with greater depth to diameter ratio (**a**) and the relevant SEM-EDS line scanning in it (**b**), the relative content of Al, Nb and Ti along the dot line marked in (**b**) by SEM-EDS line scanning: (**c**,**d**) for Al, (**e**) and (**f**) for Nb, (**g**) and (**h**) for Ti.

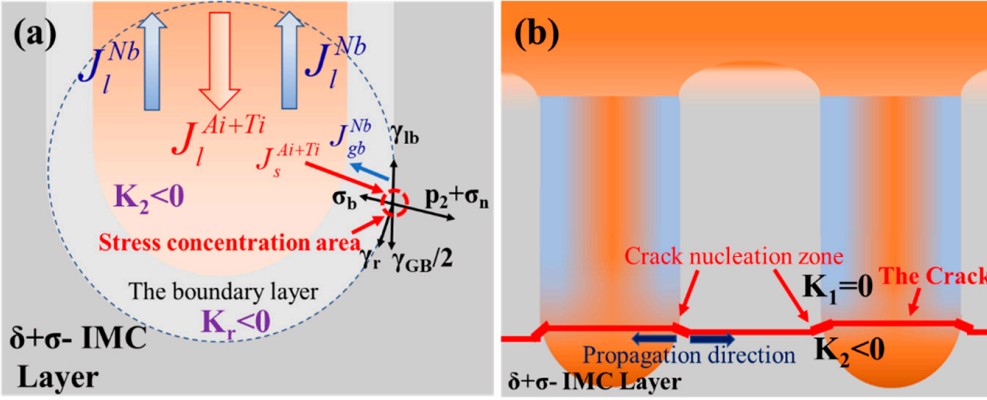

**Figure 14.** (**a**) Schematic illustration of the related stress interactions coupled with element diffusion, (**b**) schematic illustration of crack nucleation and propagation.

Experimentally, the nucleation and extension of the crack can be indirectly analyzed coupling with the GB diffusion, which is consistent with theoretical analysis in Section 4.1. It is concluded that the cracks nucleate at the boundary and then propagate to both sides to form transgranular cracks whatever the morphology of the grooves evolves.

### 4.3. Passivation and Crack Prediction

As discussed in the previous section, the stress concentrates at the bottom of the U-groove finally when the curvature $K_r$ reducing, leading to the U-groove root is passivated. However, $J_s^{Ai+Ti} > 0$ increases sharply with $K_2$ increasing rapidly, coupling with a large $P_2$ based on the analysis of Figure 14. As a consequence, the groove shape will be changed, especially, at the bottom of the groove. This will cause the location of the stress concentration area to change, namely the crack nucleation area and extension path.

For a long corrosion, however, the curvature $K_r$ of groove root continues to decrease. Based on Equation (27), $K_r$ may become too small to be neglected, resulting in a passivated U-groove tip on account of the grooves have great difficulty in grooving into the materials and from the flat. The dissolution rate of the δ+σ-IMC layer can be described by Nernst-Einstein relation [64]:

$$v = -\frac{D_s}{kT} \cdot \frac{\partial \mu}{\partial s} = -\frac{D_s \gamma \Omega}{kT} \frac{\partial K}{\partial s} \tag{30}$$

Based on Equation (30), the dissolution rate of the δ+σ-IMC layer is significantly reduced, which indicates the groove tip is passivated again. Additionally, this situation may cause the grain boundary pinning. Eventually, a steady state will be reached and the penetration rate will be controlled by the slower diffusing species. Therefore, it can be inferred that the two new wedges formed at the U-groove root under the condition of $J_s^{Ai+Ti} > 0$ coupling with $P + \sigma_n$ when $K_r \approx 0$, establishing a stress concentration zone between the new grooves, as shown in Figure 15a. In this case, the concentrated forces $P + \sigma_n$ directs outwards at the ends of the new V-groove root, which approves approximately crack propagation by the insertion of a wedge. Thermodynamically, the variation of the energy balance for crack extension in this short section of the GB will permit the crack to spread into this region [65]. It is demonstrated that the concentrated local stress may lead to preferred nucleation and accelerated growth of crack. This inference supports the experimental phenomenon perfectly, as portrayed in Figure 2e. Which is accordance with void nucleation in steel [66]. It is concluded that the growth of the V-groove is driven by reducing the curvature of U-groove root edge profile. The crack nucleates at the interfacial stress concentration area and multiplies along both sides of the interface, as presented in Figure 15b. we can draw a conclusion that the change of curvature in grooves has a very important influence on the initiation of grain boundary cracks. Consequently, the synergy of GB diffusion and stress concentration pave a way to a near surface crack and it propagates in a transgranular cracking manner, especially, when the environmental corrosion is notable.

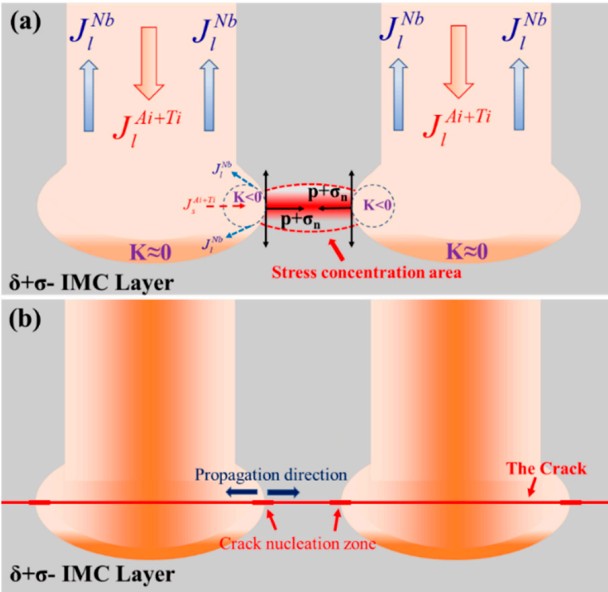

**Figure 15.** Stress interaction near the blunted U-groove tip and crack propagation: (**a**) the related stress interactions coupled with element diffusion, (**b**) schematic illustration of crack nucleation and propagation.

## 5. Crack Mechanism Map

By analyzing the interaction of the forces coupled with the GBW and GBD (containing the curvature variation of the U-groove tip and the concentration change near the flat wall of it.) in previous section, we can acquire a crack mechanism map, as shown in Figure 16.

It can be seen from the map that three main crack zones can be obtained, namely 0.63 L zone (A zone in Figure 16), 0.82 L zone (B zone in Figure 16) and the bottom crack zone (including C, D and E zones in Figure 16), and the location of crack nucleation depends on the variable curvatures and the relative intensity of GBW and GBD. In A zone, owing to an unstable curvature existed at GB, the wetting angle decreases sharply (the hollow asterisks in Figure 16) with the increase of corrosion time coupling with $\gamma_{gb} > 2\gamma_s$, resulting in the crack is preferentially initiated at the interface of 0.63 depth of the V-groove and the main controlling factors for the cracks is GBW. Then entering the complex B crack zone, the occurrence of cracks is mainly induced by GBW and GBD. As the wetting angle at the bottom of the newly formed U-groove increases (the solid asterisks in Figure 16), the curvature $K_2$ also increases, the crack area is located in 0.82 depth of the wetting channels and the cracks controlling factor will transfer from GBW to GBD. Compared with A and B zones, the generation of cracks is mainly controlled by GBD in C, D and E zones. By contrast, the generation of the crack in the C zone is still affected by a weakened GBW owing to the curvature of the U-groove root ($K_r$) and wall ($K_1$) decreases as the wetting angle increases and the crack nucleation area is located in the boundary layer region of $K = 0$ between $K_1$ and $K_2$, while in D zone the crack is almost unaffected by GBW because of the wetting angle exceeds the theoretical maximum value, and the crack initiates at GB where $K_2 < 0$. As the wetting angle exceeds 150°, it seems to the effect of dewetting appears instead of GBW, the crack nucleation is at the bottom of U-groove where $K_2 < 0$ and is only affected by GBD. As per Figure 16, it can be concluded that the change in the curvature variation of the U-groove root caused by GBD has a very important effect on the nucleation of the crack. Consequently, based on the crack mechanism map, the crack nucleation position and the main factors affecting crack nucleation can be inductively obtained and distinguished. This will play a significant role in understanding and exploiting the GBW and GBD in engineering processes, such as welding and evaporation of liquid. A low degree of GBD and GBW (the wetting angle > 90°) are required in welding process as shown in yellow part of A zone in Figure 16 where a short interaction time is demonstrated, otherwise, welding cracks will occur. Another example is evaporation of liquid in the GBs in which the liquid should be de-wetted (the wetting angle > 140°) from GBs, a long time and a big GBD are required as shown in E zone in Figure 16.

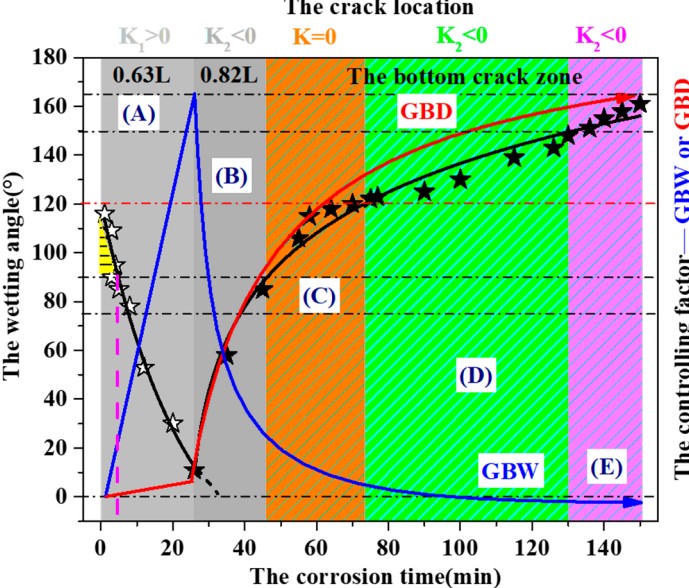

**Figure 16.** The proposed crack mechanism map with the dependence on GBW and GBD, and the related crack nucleation area, A and B zone are mainly affected by GBW while C, D and E zones mainly rely on GBD. (A: 0.63 L zone, B: 0.82 L zone, C, D and E zones: the bottom crack zone.).

## 6. Summary

This study proposes an approach to investigate self-induced internal corrosion stress transgranular cracking by synergize GBW and GBD, which is very distinguished from the general SCC. The dynamic self-induced internal corrosion stress is indirectly analyzed coupling with diffusion flux theoretically and experimentally. The transgranular failure mechanism has been discovered for the material exposed to high temperature melt corrosion, and two types of crack initiation modes are recognized and distinguished as V-grooves and U-grooves fracture. The experimental results designate a tendency that self-induced internal corrosion stresses interact at GB interface to form a stress concentration zone where the crack initiates preferentially, then the crack propagates along both sides of the interface and traverses the columnar IMC grains. The source of stresses is mainly caused by GBW and the interface curvature variation in the V-grooves. In contrast, the stresses largely coupled with the GB diffusion and variation of the interface curvature in the U-grooves. The corrosion stress transgranular cracking studied here may provide some new insights into the fracture mechanisms, a mechanism map is established to reveal crack nucleation and growth with synergy of GBW, GBD and interface curvature variation. The projected framework will be protracted to a grain boundary wetting and diffusion engineering.

**Author Contributions:** Conceptualization, F.K.; methodology, X.L.; formal analysis, X.L.; investigation, X.L.; re-sources, X.W.; data curation, X.L.; writing—original draft preparation, X.L.; writing—review and editing, X.L.; supervision, Y.C.; project administration, F.K.; funding acquisition, H.Z. All authors have read and agreed to the published version of the manuscript.

**Funding:** This study was financially supported by Shanghai Sailing Program, grant number 19YF1420000.

**Data Availability Statement:** Not applicable.

**Conflicts of Interest:** The authors declare no conflict of interest.

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
