# Peer review of "Self-Induced Internal Corrosion Stress Transgranular Cracking in Gradient-Structural Ploycrystalline Materials at High Temperature"

_metals, doi:10.3390/met11091465_

Round 1

Reviewer 1 Report

First of all, extensive English editing is required. It is extremely hard to follow the discussion, so the reviewing process cannot be properly done. Some important statements could have been misinterpreted so it is strongly recommended to submit a new version with a fluid writing.

Regarding the technical content, the topic is appealing for the scientific community focused on environmental degradation phenomena. Advances in this research line would surely facilitate grain boundary engineering and so a better performance of materials and alloys. The proposal of a “crack mechanism map” including grain boundary wetting and diffusion is appreciated. However, I do not understand that the authors refer to “self-induced internal corrosion stress” since the stress is produced in the interphase between a TiAl melt and a Nb container and this is a problem of inter-diffusion of elements, curvature-dependent stress and stress-driven diffusion. Additionally, the authors generally speak about “corrosion” without any explanation of electrochemical reactions or mechanisms. I have the feeling that the authors successfully explain mechanisms but the background in some concepts is weak.  

Little attention is given to the formation of grooves, which is crucial for understanding this mode of failure. The objective of analysing the curvature and the groove shape effects on diffusion is promising and one of the strong points of this manuscript. The correlation between atoms diffusing through the GB and cracking is plausible. However, the theoretical elaboration of diffusion phenomena is very poor with many typos in equations and little insight in stress-assisted diffusion.

Experimental methods are adequate, including SEM-EDS and EPMA techniques that facilitate the identification of elements and cracking features.

Therefore, I recommend the major revision of this manuscript, putting the focus on English editing, a more clear elaboration of diffusion governing equations and a better discussion.

The following concerns should also be addressed:

  • I am not very sure about the statement “LME is mainly focused on the corrosive environment of liquid metals”. LME is a complex process but from my point of view it all proposed mechanisms invoke absorption and diffusion but redox phenomena take a secondary role, so it is confusing to talk about corrosive environments.
  • I do not understand the meaning of the sentence: “the normal stresses were adjacent to the composition during GB interdiffusion process”. Could you clarify this?
  • The phases epsilon, delta and beta are not defined.
  • The zones (A) to (E) in the cracking mechanism map (Fig 15) should be also briefly defined in the caption.
  • The analysis of fracture morphology should be, in my opinion, placed before discussion. The appearance of cleavage is a starting point, not a final verification of transgranular fracture. The identification of the zone where these micrographs have been taken would be appreciated.

Author Response

Appreciation for your review comments to our manuscript again. We acknowledge that there are a lot of mistakes in the revised manuscript, which compromise the quality of our manuscript. In response to your comments, we have revised the manuscript in details. We hope, with these modifications and improvements, the quality of the English would meet the publication standard of Metals.

The revisions have been done in the attached manuscript and conducted in the paper as the highlight text (the dark red color). Some explanations, argument, and actions regarding the revisions of our manuscript are as follows.

Reviewer1

First of all, extensive English editing is required. It is extremely hard to follow the discussion, so the reviewing process cannot be properly done. Some important statements could have been misinterpreted so it is strongly recommended to submit a new version with a fluid writing.

Answer and action:

I have this paper corrected for English by a professional American teacher firstly, then I have revised the manuscript carefully (the red color), hoping the quality of the English would meet the publication standard of Metals.

Regarding the technical content, the topic is appealing for the scientific community focused on environmental degradation phenomena. Advances in this research line would surely facilitate grain boundary engineering and so a better performance of materials and alloys. The proposal of a “crack mechanism map” including grain boundary wetting and diffusion is appreciated. However, I do not understand that the authors refer to “self-induced internal corrosion stress” since the stress is produced in the interphase between a TiAl melt and a Nb container and this is a problem of inter-diffusion of elements, curvature-dependent stress and stress-driven diffusion. Additionally, the authors generally speak about “corrosion” without any explanation of electrochemical reactions or mechanisms. I have the feeling that the authors successfully explain mechanisms but the background in some concepts is weak.

Explanation and argument:

The intergranular failure is usually known as liquid metal embrittlement (LME) and stress corrosion cracking (SCC). The phenomenon of LME may be defined as the brittle fracture, or loss in ductility, of a usually ductile material in presence of liquid metal [1], while SCC is developed from simple metallurgical principles and assuming that high surface mobility is present in the process[2]. After we discussed, it is demonstrated that the corrosion in our manuscript is SCC owing to a metallurgical interaction between the TiAl melt and Nb container at high temperature, especially, the interaction between intermetallic compound layer covered Nb container and TiAl melt. In this process, the change of stress is related to the evolution of interface morphology caused by the spontaneous diffusion (caused by the change of curvature and the concentration of the relative element), which is “self-induced internal corrosion stress”. The related corrosion mechanism is the Nb container was corroded interfacial interaction caused by inter-diffusion of elements, curvature-dependent stress and stress-driven diffusion.

Little attention is given to the formation of grooves, which is crucial for understanding this mode of failure. The objective of analysing the curvature and the groove shape effects on diffusion is promising and one of the strong points of this manuscript. The correlation between atoms diffusing through the GB and cracking is plausible. However, the theoretical elaboration of diffusion phenomena is very poor with many typos in equations and little insight in stress-assisted diffusion.

Explanation and action:

The formation and evolution of grooves had been studied in our other research paper[3], and we quote in reference [52].

The purpose of the theoretical elaboration of diffusion phenomena is to establish internal corrosion stress is mainly affected by chemical potential gradient, interface curvature variation, and interfacial diffusion, paving the way for the following analysis.

Experimental methods are adequate, including SEM-EDS and EPMA techniques that facilitate the identification of elements and cracking features.

Therefore, I recommend the major revision of this manuscript, putting the focus on English editing, a more clear elaboration of diffusion governing equations and a better discussion.

The following concerns should also be addressed:

  • I am not very sure about the statement “LME is mainly focused on the corrosive environment of liquid metals”. LME is a complex process but from my point of view it all proposed mechanisms invoke absorption and diffusion but redox phenomena take a secondary role, so it is confusing to talk about corrosive environments.

Answer and action:

This sentence is a misrepresentation in the manuscript. We have deleted it.

  • I do not understand the meaning of the sentence: “the normal stresses were adjacent to the composition during GB interdiffusion process”. Could you clarify this?

Answer and action:

This sentence should be “the normal stress will be generated during the GBD process resulting from the dependence of grain boundary energy on composition.”

We have corrected it in the manuscript.

  • The phases epsilon, delta and beta are not defined.

Answer and action:

The phases epsilon, delta and beta have been defined in Table 2.

Table 2 the phases and the correlative chemical formula

Phase

Chemistry formula

δ

Nb3Al

σ

Nb2Al

ε

(Ti1-xNbx)Al3, TiAl3(h), NbAl3

Ο2

Ti2NbAl

β

(Nb)ss

  • The zones (A) to (E) in the cracking mechanism map (Fig 15) should be also briefly defined in the caption.

Answer and action:

The zones (A) to (E) have been defined in the caption.

Figure 16. The proposed crack mechanism map with the dependence on GBW and GBD, and the related crack nucleation area, A and B zone are mainly affected by GBW while C, D, and E zones mainly rely on GBD. (A: 0.63L zone, B: 0.82L zone, C, D, and E zones: the bottom crack zone.)

  • The analysis of fracture morphology should be, in my opinion, placed before discussion. The appearance of cleavage is a starting point, not a final verification of transgranular fracture. The identification of the zone where these micrographs have been taken would be appreciated.

Answer and action:

The fracture morphology part has been moved to section 3.

[1] A. Turnbull Stress Corrosion Cracking in Metals – Mechanisms. Reference Module in Materials Science and Materials Engineering. Elsevier, 2016.

[2] J. Galvele, A stress corrosion cracking mechanism based on surface mobility, Corrosion Science, (1987) 27: 1-33.

[3] X. Lei, X. Wang, F. Kong, Y. Chen, The high temperature wetting and corrosion mechanism analysis of Nb by TiAl alloy melt, Corrosion Science, (2021) 186: 109316.

Reviewer 2 Report

The manuscript needs correction:

  • In the annotation, keywords and output, it is necessary to add the grades of the materials of the container and the titanium alloy under study;
  • be sure to detail or leave the main literary sources instead of the indicated ranges (see [24, 30-38]. On page 2 and [40, 62-64] on page 5);
  • indicate the source of the chemical composition of the starting materials in table 1: certificate or equipment;
  • be sure to provide a method for determining the chemical composition of the melted titanium alloy and statistical evidence of the homogeneity of the samples obtained;
  • it is desirable to show photographs of the original containers and samples, as well as the process of their placement or removal from the oven chamber;
  • clarify the melting plant. If the installation is standard, give the brand, if specially designed, then indicate the details of its control system (types of sensors, measurement accuracy of controlled parameters, etc.). Be sure to add argon to the circuit in Figure 1;
  • the introductory paragraph of Section 4 should be moved to the overview chapter and clearly formulated the purpose of the research. In the current version of the manuscript, it is blurry;
  • exclude the re-introduction of Figure 13 into the text (now Figure 13 and Figure 14 are the same);
  • in the map of the mechanism of cracks (Fig. 15) decipher the "asterisks";
  • in the output and annotation, it is imperative to reflect the map of the fracture mechanism, since it is of great importance to engineering practice.

Author Response

Appreciation for your review comments to our manuscript again. We acknowledge that there are a lot of mistakes in the revised manuscript, which compromise the quality of our manuscript. In response to your comments, we have revised the manuscript in details. We hope, with these modifications and improvements, the quality of the English would meet the publication standard of Metals.

The revisions have been done in the attached manuscript and conducted in the paper as the highlight text (the dark red color). Some explanations, argument, and actions regarding the revisions of our manuscript are as follows.

Reviewer 2

The manuscript needs correction:

  • In the annotation, keywords and output, it is necessary to add the grades of the materials of the container and the titanium alloy under study;

Answer and action:

The relative information of materials has been added.

Table 1. Component and form of materials used in experiments.

material

Form

Chemical composition (wt%)

Ti (bala.)

sponge

Fe

Si

Cl

C

N

O

Mn

Mg

H

0.015

0.009

0.047

0.009

0.005

0.047

0.003

0.004

0.001

Al (bala.)

Lump

Fe

Si

Ga

Cu

Mg

Zn

0.1

0.03

0.02

0

0

0.01

Nba (bala.)

Tube

C

N

H

O

Fe

Si

Mo

W

Ti

Ta

Ni

Hf

Zr

8

49

3

110

15

30

32

140

<5

500

<5

20

32

a Chemical composition ≤(ppm wt)

Table 2 the phases and the correlative chemical formula

Phase

Chemistry formula

δ

Nb3Al

σ

Nb2Al

ε

(Ti1-xNbx)Al3, TiAl3(h), NbAl3

Ο2

Ti2NbAl

β

(Nb)ss

  • be sure to detail or leave the main literary sources instead of the indicated ranges (see [24, 30-38]. On page 2 and [40, 62-64] on page 5);

Answer and action

The main literary sources have been retained, and deleted some unimportant references.

  • indicate the source of the chemical composition of the starting materials in table 1: certificate or equipment;

Answer and action:

The source of the chemical composition of the starting materials in table 1 from three certificates.

Table 1. Component and form of materials used in experiments.

material

Form

Chemical composition (wt%)

Ti (bala.)

sponge

Fe

Si

Cl

C

N

O

Mn

Mg

H

0.015

0.009

0.047

0.009

0.005

0.047

0.003

0.004

0.001

Al(bala.)

Lump

Fe

Si

Ga

Cu

Mg

Zn

0.1

0.03

0.02

0

0

0.01

Nba (bala.)

Tube

C

N

H

O

Fe

Si

Mo

W

Ti

Ta

Ni

Hf

Zr

8

49

3

110

15

30

32

140

<5

500

<5

20

32

a Chemical composition ≤(ppm wt)

  • be sure to provide a method for determining the chemical composition of the melted titanium alloy and statistical evidence of the homogeneity of the samples obtained;

Answer and action:

The cast ingot had a chemical composition of Ti-47.05Al (at.%) by EPMA analysis.

Four samples were randomly cut from the ingot for EDS analysis, and the chemical composition were as follows.

Point

Ti

Al

1

52.8

47.2

2

53.2

46.8

3

52.8

47.2

4

53

47.0

Average

52.95

47.05

  • it is desirable to show photographs of the original containers and samples, as well as the process of their placement or removal from the oven chamber;

Answer and action:

The pictures of materials and sketch of experimental apparatus have been shown below.

Figure 1. Sketch of experimental apparatus [52].

  • clarify the melting plant. If the installation is standard, give the brand, if specially designed, then indicate the details of its control system (types of sensors, measurement accuracy of controlled parameters, etc.). Be sure to add argon to the circuit in Figure 1;

Answer and action:

The controlled parameters have been added in section 2, and the related gas circuit has been also added in figure 1.

Figure 1. Sketch of experimental apparatus [52].

  • the introductory paragraph of Section 4 should be moved to the overview chapter and clearly formulated the purpose of the research. In the current version of the manuscript, it is blurry;

Answer and action:

The introductory paragraph of Section 4 has been moved to the overview chapter and clearly formulated the purpose of the research.

  • exclude the re-introduction of Figure 13 into the text (now Figure 13 and Figure 14 are the same);

Answer and argument:

Figure 13 and Figure 14 are different even though passivation occurs in both cases.

Firstly, the stress is produced in the interphase between the TiAl melt and the intermetallic compound layer and this is a problem of stress coupled with diffusion, which includes inter-diffusion of elements, curvature of the grooves. The grooves in Figure 13 and Figure 14 have different curvature, therefore, they are different situations.

Secondly, there is a crucial information, namely, the change of wetting and dewetting, which corresponds to the two regions (D and E zones) in the crack mechanism map.

Based on the above two points, we insist on separating the two pictures.

  • in the map of the mechanism of cracks (Fig. 15) decipher the "asterisks";

Answer and action:

The "asterisks" represent the angle of the measured wetting angle and have been deciphered in the section 5. The decreasing wetting angles (the hollow asterisks) and the increasing wetting angles (the solid asterisks).

  • in the output and annotation, it is imperative to reflect the map of the fracture mechanism, since it is of great importance to engineering practice.

Answer and action:

We have added the annotation to reflect the map of the fracture mechanism below the map.

Figure 15. The proposed crack mechanism map with the dependence on GBW and GBD, and the related crack nucleation area, A and B zone are mainly affected by GBW while C, D, and E zones mainly rely on GBD. (A: 0.63L zone, B: 0.82L zone, C, D, and E zones: the bottom crack zone.)

Round 2

Reviewer 2 Report

Good. But the crack mechanism map should be mentioned in the conclusions and annotations.

Author Response

Appreciation for your review comments to our manuscript again. We acknowledge that there are a lot of mistakes in the revised manuscript, which compromise the quality of our manuscript. In response to your comments, we have revised the manuscript in details. We hope, with these modifications and improvements, the quality of the English would meet the publication standard of Metals.

The revisions have been done in the attached manuscript and conducted in the paper as the highlight text (the green color). Some explanations and actions regarding the revisions of our manuscript are as follows.

Reviewer 2#

Good. But the crack mechanism map should be mentioned in the conclusions and annotations.

Answer and action:

The crack mechanism map has been mentioned in the conclusions and annotations (in section 1, section 5, and section 6).
